# Hot Deformation Characteristics and Processing Parameter Optimization of Al–6.32Zn–2.10Mg Alloy Using Constitutive Equation and Processing Map

**Zhengbing Xiao** [1,2], **Qiang Wang** [1], **Yuanchun Huang** [1,*], **Jiawei Hu** [1] and **Ming Li** [1]

[1] Light Alloy Research Institute, Central South University, Changsha 410083, China; xiaozb@csu.edu.cn (Z.X.); 183812053@csu.edu.cn (Q.W.); hujwxy@csu.edu.cn (J.H.); csuliming@csu.edu.cn (M.L.)

[2] State Key Laboratory of High Performance Complex Manufacturing, Central South University, Changsha 410083, China

\* Correspondence: science@csu.edu.cn; Tel.: +86-135-0731-5123; Fax: +86-188-877-856

**Abstract:** Hot compression tests over the temperature range from 350 °C to 500 °C and strain rates range from 0.001 s$^{-1}$ to 1 s$^{-1}$ for homogenized Al–6.32Zn–2.10Mg alloy were carried out on a Gleeble-3800 thermal simulation machine to characterize its hot deformation behavior. At the same time, a modified Arrhenius constitutive equation was established to describe the flow behavior of the alloy, whose average absolute error is 2.89%, which proved to have an excellent predictive effect on the flow stress of the alloy. The hot processing map of the alloy was established, and the stability processing parameters were 460–500 °C and 0.01–0.08 s$^{-1}$. Then, the Z parameter processing map and activation energy processing (AEP) maps were established for further optimization. Eventually, the optimal processing parameters of the alloy was 460–500 °C (0.03–0.08 s$^{-1}$). Then, the microstructure of specimens was observed using electron backscatter diffraction. Based on the findings the reasonability of the AEP map and Z parameter map was verified. Finally, electron backscatter diffraction (EBSD) techniques were used to analyze the evolution of the grain structure during the deformation process. It was found that dynamic recovery (DRV) was the main softening mechanism of Al–6.32Zn–2.10Mg. Continuous dynamic recrystallization (CDRX) and discontinuous dynamic recrystallization (DDRX) operated together with the increase of strain, but CDRX was confirmed as the dominant DRX mechanism.

**Keywords:** Al–6.32Zn–2.10Mg alloy; Hot deformation; Hot processing map; AEP map; microstructure characteristics



## 1. Introduction

Al–Zn–Mg alloys play an integral role in the aerospace industry and other defense industries as one of the main structural materials owing to their high strength and good toughness. Unfortunately, various defects such as voids and fractures are difficult to avoid during hot processing, especially when the processing parameters are not selected properly [1,2]. Therefore, it is essential to reveal the relationship between the flow behavior and microstructure evolution of hot deformation, to study the true stress–strain curves of the alloy, and to establish the high temperature rheological constitutive model and hot processing map, which is of great significance to optimize the alloy production process and improve the product performance and quality.

In research on the metal flow behavior, the high temperature rheological constitutive model is used to predict the true stress of alloys under different deformation parameters and is the key to research into its hot deformation behavior and carrying out the numerical simulation. The constitutive model can be referred to as a macroscopic phenomenological constitutive model, microscopic physical constitutive model and artificial neural network (ANN) constitutive model [3–5]. In past research, it has been proved that the Arrhenius

constitutive equation theory is a feasible method to investigate the relationship among deformation temperature, strain rate and true stress during hot deformation of alloys, The Arrhenius constitutive equation is established based on the regression method, and the relationship between the constitutive equation and the deformation mechanism is less strictly connected, so it has been widely used in practical applications and has always had a good performance in establishing the relationship between deformation parameters. [6–8]. In the hot forming process of Al–Zn–Mg alloys, it is significant to select reasonable hot-processing parameters for obtaining the ideal component microstructure and avoiding defects such as intergranular cracks, adiabatic shear bands, and the generation and propagation of microcracks. The hot processing map is an important basis for judging the suitable processing area and unsuitable processing area of the alloy by superimposing the power dissipation map and the instability map. It can reflect the change mechanism of the internal structure of the material during hot deformation, and evaluate the process ability of the material to determine the "safe area" and "instability area". However, the stability of deformation is considered in conventional hot-processing maps, but overlooks the difficulty of deformation, and it is difficult for hot processing maps to efficiently and accurately obtain the optimal processing parameters. In the meantime, the activation energy represents the ability of atoms to overcome energy barriers during deformation and is often used to characterize the difficulty of deformation. Hence, it is necessary to introduce an activation energy processing (AEP) map to further optimize the suitable processing parameters obtained from a hot-processing map. Finally, through the coupling of a hot-processing map and AEP map, an optimal processing parameter with high dissipation coefficient and low activation energy is obtained. Moreover, the microstructure of hot deformation can be controlled by the hot processing map and AEP map [9,10]. Therefore, it is very important to research the coupling relationship between the hot-processing map and AEP map microstructure.

To enhance the overall properties of the Al–Zn–Mg alloys, a great number of studies have focused on the compositional variables and hot-processing map of the alloys [11–13]. The precipitation phase morphology and precipitation phase of alloys vary by composition content, which will directly affect the hot deformation behavior of the alloy, thereby affecting the hot-processing map of the alloy [14–16]. Sun et al. [17] investigated the effects of deformation conditions on diffusion activation energy $Q$ and the influences of the $Z$ parameter on microstructural evolution and deformation mechanism of the Al–Zn–Mg alloy, However, the Z parameter and the activation energy $Q$ value are not coupled with the processing map to optimize the process parameters; Wu et al. [18] established the processing map under different strains of a new type Al–Zn–Mg–Er–Zr alloy, and determined the range of flow instability and the best thermal processing parameters. Lin et al. [19] established the hot-processing map of Al–5.8Zn–2.3Mg by superimposing the power dissipation coefficient diagram and the instability diagram, and studied the relationship between microstructure evolution and hot working. Nevertheless, this was not only not through the coupling of the hot processing map and AEP map to optimal processing parameter, but also failed to explore more details about the evolution of the grain structure during the hot deformation.

As mentioned above, both process parameters and chemical composition are critical factors to determine the hot deformation behavior of metals. However, no such systematic research on the hot deformation behavior of Al–6.32Zn–2.10Mg alloy has appeared. No study has examined the coupling relationship between hot processing map and AEP map of Al–Zn–Mg alloy. Therefore, in the present work on the alloy after homogenizing was applied, the hot compression test was performed at isothermal constant strain rate on a Gleeble-3800 thermo-simulation machine aiming to establish a modified Arrhenius model and draw a hot-processing map and AEP map. The mapping relations between microstructure and high temperature deformation parameters of the alloy was characterized, which supplied a theoretical basis for optimizing the hot working processes of the alloy and improving the performance and quality of the hot-processed products.

## 2. Experimental

As-cast Al–Zn–Mg alloys after homogenizing were used in the experiment, whose composition is shown in Table 1. Then the alloy was machined into a cylindrical rod specimen with 10 mm in diameter and 15 mm in height. The surface of the circular column specimens is sanded with 800 sandpaper to eliminate surface defects.

**Table 1.** Chemical compositions of the studied alloy (wt.%).

| Chemical Composition | Zn | Mg | Cu | Zr | Ti | Fe | Si | Al |
|---|---|---|---|---|---|---|---|---|
| wt.% | 6.32 | 2.10 | 0.10 | 0.11 | 0.063 | 0.063 | 0.038 | Bal. |

The hot compression experiments were carried out on a Gleeble-3800 thermo-simulation machine. Both ends of the specimens were coated with lubricant to decrease the influences of friction on the stress state. As shown in Figure 1, during the hot compression test, the specimens were heated to the set deformation temperature to 350, 400, 450 and 500 °C at a rate of 5 °C/s, and kept for 5 min to ensure that the internal temperature of the specimen was uniform. Following that, isothermal compression deformation was performed at specific strain rates of 0.001, 0.01, 0.1, and 1 $s^{-1}$. The compression reduction for the specimens was set as 60%, i.e., true strain was about 0.9. They were quenched in water rapidly after the end of compression to retain their thermally deformed structure. The deformed samples were cut in half parallel to the direction of the compression direction and the tissue in the center of the sample was observed. The hot deformation microstructure was observed by electron backscatter diffraction (EBSD, Mechanical polishing is performed on the sample first, and then electrolytic polishing was performed on the longitudinal section in perchloric acid solution ($HClO_4$ 10 mL + ethanol 90 mL) with 20 v power supply for 3–5 s to remove stress and surface micro-marks.).

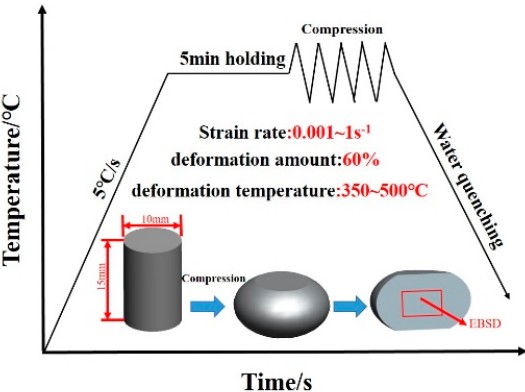

**Figure 1.** Hot-deformation process of alloy.

## 3. Results

### 3.1. True Stress–True Strain Curves

The true stress–strain curves of the Al–6.32Zn–2.10Mg alloy under different deformation parameters are shown in Figure 2. The true stress-strain curves of the alloy can be obviously divided into three parts: the work hardening stage, the dynamic recovery stage, and the steady state roughly. There is an apparent trend in all the cases of the true stress of the alloy reducing with the increase of temperature and the decrease of strain rate. This is because the lower strain rate and the higher temperature lead to the longer energy accumulation time and the greater dynamic softening driving force, which is beneficial for dynamic softening to eliminate dislocation entanglement. At the same time, it can be obtained from the deformation curve that after reaching the peak stress during deformation, the flow curve hardly drops. Only when the temperature is 400–500 °C and low strain rates

($0.001 \text{ s}^{-1}$ and $0.01 \text{ s}^{-1}$), a small single peak appears before reaching the steady state, and the wave feature can be considered as the appearance of dynamic recrystallization [20].

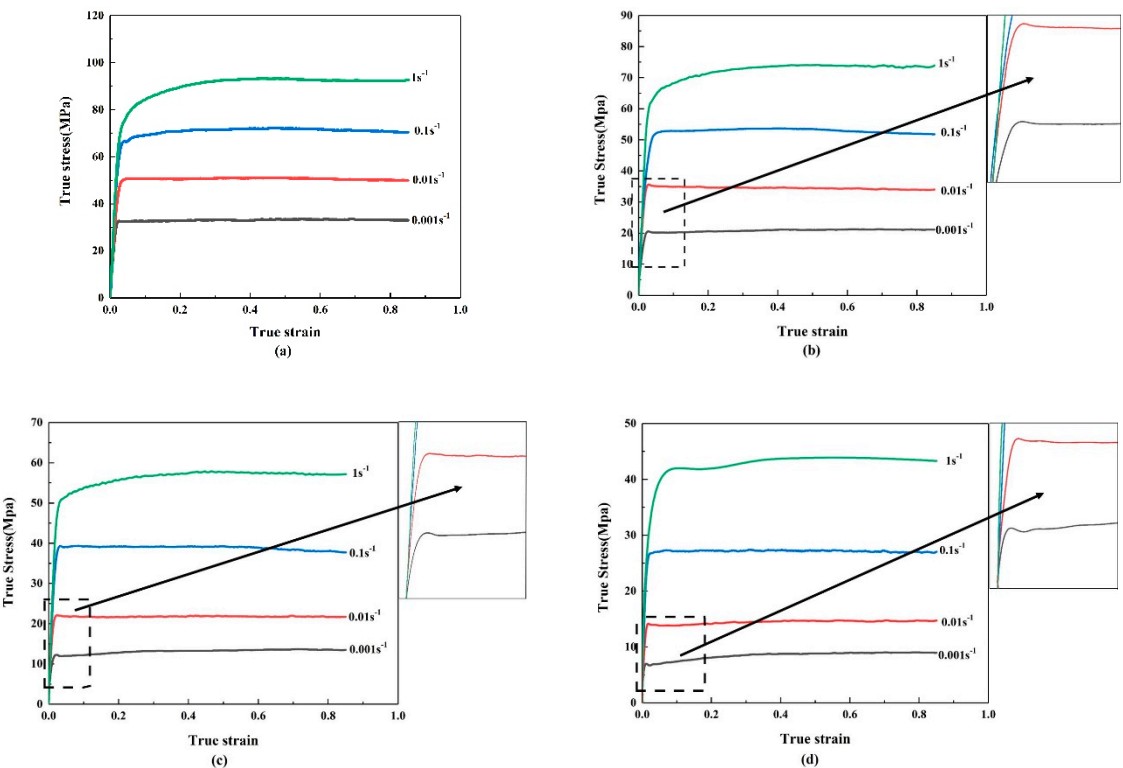

**Figure 2.** True stress–true strain curves of alloy: (**a**) 350 °C; (**b**) 400 °C; (**c**) 450 °C; (**d**) 500 °C.

### 3.2. Modified Arrhenius Model

The Arrhenius equation can not only be used predicate to express the influences of temperature and strain rate on high-temperature flow stress, but also to obtain the hot deformation activation energy of the alloy [21,22], which can be expressed as follows preference:

$$\dot{\varepsilon} = AF(\sigma)\exp\left(-\frac{Q}{RT}\right) \tag{1}$$

where $\dot{\varepsilon}$ ($\text{s}^{-1}$) is the strain rate, $\sigma$ (MPa) is the true stress during hot deformation, $Q$ (kJ/mol) is the hot deformation activation energy, and $R$ ($8.3145 \text{ J} * \text{mol}^{-1} * \text{K}^{-1}$) is gaseous constant, A is the material constant, and $F(\sigma)$ in the formula can be expressed as below in terms of in terms of the magnitude of stress preference [7,8]:

$$F(\sigma) = \sigma^N (\alpha\sigma \leq 0.8) \tag{2}$$

$$F(\sigma) = \exp(\beta\sigma)(\alpha\sigma \geq 1.2) \tag{3}$$

$$F(\sigma) = [\sinh(\alpha\sigma)]^n (\text{For all } \sigma) \tag{4}$$

where $N$, $\beta$, $n$, $\alpha$ are material constants and $\alpha$ can be expressed as $\alpha = \beta/N$. Moreover, others can be obtained according to the Equations (5) and (6) below, which are the logarithm of Equations (2) and (3) for both sides:

$$\ln\dot{\varepsilon} = \ln A_1 + N\ln\sigma - \frac{Q}{RT}(\alpha\sigma \leq 0.8) \tag{5}$$

$$\ln\dot{\varepsilon} = \ln A_2 + \beta\sigma - \frac{Q}{RT}(\alpha\sigma \geq 1.2) \tag{6}$$

To use Equation (5) to calculate out $N$, stress out $\varepsilon = 0.1$ were selected out of consideration of the applicable range Equation (5) ($\alpha\sigma \leq 0.8$), constants $N = 5.837$ are calculated from the average values of the slopes of $\ln\dot{\varepsilon} - \ln\sigma$ plots in the deformation temperature range between 350 and 500 °C. Similarly take strain $\varepsilon = 0.4$. At this time, the corresponding stress value is near the peak stress. β can be obtained from the $\ln\dot{\varepsilon} - \sigma$ plots as 0.160 in Figure 3b. Finally, $\alpha$ can be taken as 0.027 by deriving $\alpha = \beta/N$.

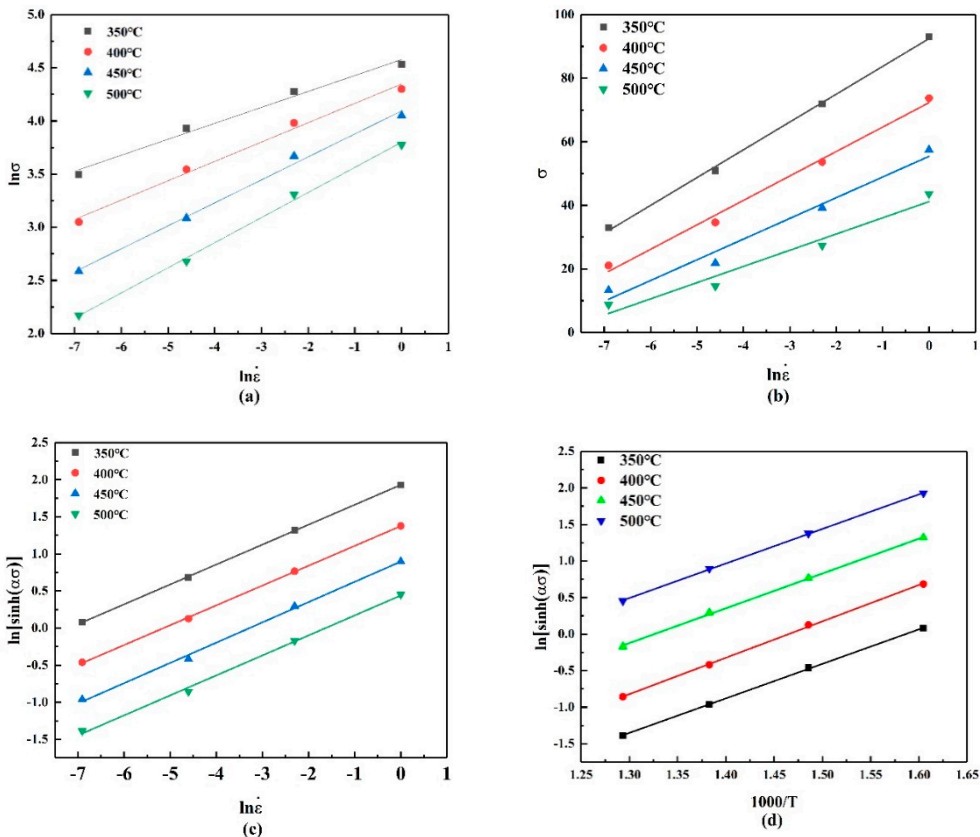

**Figure 3.** The fitting curve of the Arrhenius parameter: (**a**) $\ln\dot{\varepsilon} - \ln\sigma$ (**b**) $\ln\dot{\varepsilon} - \sigma$ (**c**) $\ln\dot{\varepsilon} - \ln[\sinh(\alpha\sigma)]$ (**d**) $\ln[\sinh(\alpha\sigma)] - 1000/T$.

Then, for all stress states, we can bring Equation (4) into Equation (1) to obtain:

$$\ln\dot{\varepsilon} = \ln A_3 + n\ln[\sinh(\alpha\sigma)] - \frac{Q}{RT} (\text{For all } \sigma) \tag{7}$$

Afterwards taking the steady-state stress value with strain $\varepsilon = 0.6$, as shown in Figure 3c, $n = 3.769$ can be obtained by linearly fitting the $\ln\dot{\varepsilon} - \ln[\sinh(\alpha\sigma)]$ curve, and Equation (7) can also be expressed as:

$$Q = R \left.\frac{\partial \ln\dot{\varepsilon}}{\partial \ln[\sinh(\alpha\sigma)]}\right|_T \left.\frac{\partial \ln[\sinh(\alpha\sigma)]}{\partial(1/T)}\right|_{\dot{\varepsilon}} = RnS \tag{8}$$

Therefore, as illustrated in Figure 3d, the $Q = 147.805$ KJ/mol is acquired by linear fitting of $\ln[\sinh(\alpha\sigma)] - 1000/T$. The $Q$ value can represent the Zener–Hollomon parameter. The Zener–Hollomon parameter can represent the effect of temperature and strain rate on the true stress [23]. Its mathematical expression is:

$$Z = \dot{\varepsilon}\exp\left(\frac{Q}{RT}\right) \tag{9}$$

Introduce Equation (9) into Equation (4) and calculate the logarithm of both sides:

$$\ln Z = n \ln[\sinh(\alpha\sigma)] + \ln A \tag{10}$$

From Equation (10), the material constants $n$ and $A$ can be available by linear fitting of the $\ln Z - \ln[\sinh(\alpha\sigma)]$ graph.

$$\sigma = \frac{1}{\alpha} \ln\left\{ \left(\frac{Z}{A}\right)^{1/n} + \left[ \left(\frac{Z}{A}\right)^{2/n} + 1 \right]^{1/2} \right\} \tag{11}$$

Remarkably, the Arrhenius model established by the above method does not consider the real strain. It is used to predict the flow stress of the alloy but is inaccurate. Consequently, the constitutive equation needs to be modified with strain compensation, to make the prediction more precise.

As displayed in Equation (12), the material constant in the modified constitutive model can be expressed as a polynomial function of strain [24]. To verify the accuracy, the data under the strain of 0.05 to 0.65 are used to calculate the material constants ($\alpha$, $Q$, $n$, and $\ln A$), and the strain increment is 0.05. The fitted curve is shown in Figure 4, and the 5th order polynomial fitting can be used to describe the strain pair. The influence of material constant, the fitted line has excellent correlation and generalization. The fitted results are listed in Table 2.

$$\begin{cases} \alpha = B_0 + B_1\varepsilon + B_2\varepsilon^2 + B_3\varepsilon^3 + B_4\varepsilon^4 + B_5\varepsilon^5 \\ Q = C_0 + C_1\varepsilon + C_2\varepsilon^2 + C_3\varepsilon^3 + C_4\varepsilon^4 + C_5\varepsilon^5 \\ n = D_0 + D_1\varepsilon + D_2\varepsilon^2 + D_3\varepsilon^3 + D_4\varepsilon^4 + D_5\varepsilon^5 \\ \ln A = F_0 + F_1\varepsilon + F_2\varepsilon^2 + F_3\varepsilon^3 + F_4\varepsilon^4 + F_5\varepsilon^5 \end{cases} \tag{12}$$

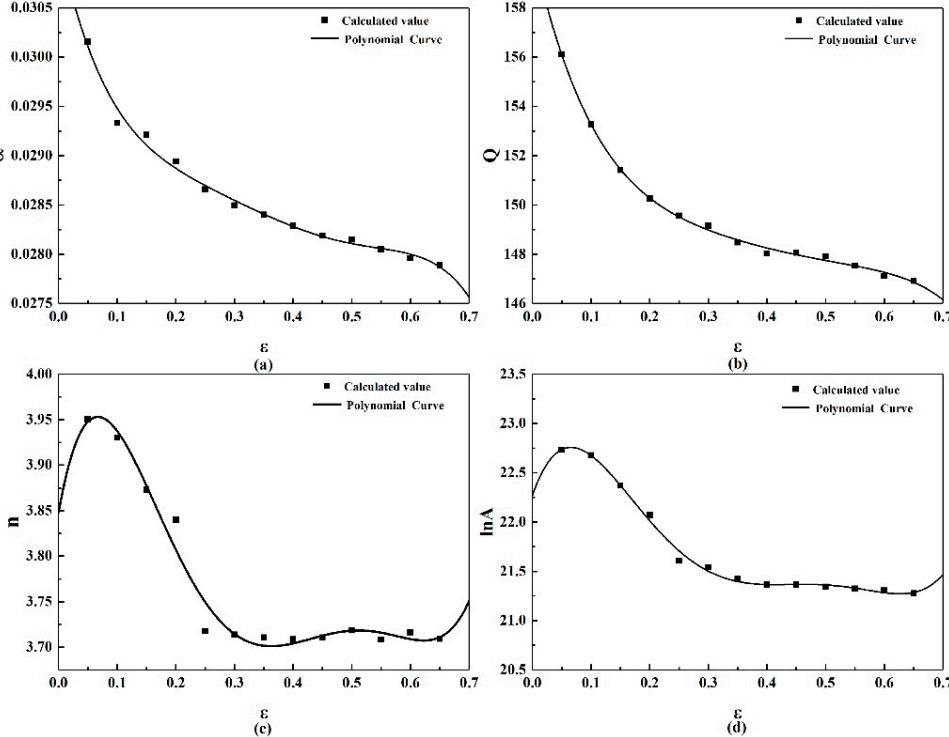

**Figure 4.** 5th order polynomial fitting: (**a**) $\alpha$ (**b**) $Q$ (**c**) $n$ (**d**) $\ln A$.

**Table 2.** Coefficients of polynomial fitting curves for material coefficients.

| $\alpha$ | $Q$ | $n$ | $\ln A$ |
|---|---|---|---|
| $B_0 = 0.031$ | $C_0 = 160.421$ | $D_0 = 3.848$ | $F_0 = 22.273$ |
| $B_1 = -0.027$ | $C_1 = -105.167$ | $D_1 = 3.613$ | $F_1 = 16.869$ |
| $B_2 = 0.133$ | $C_2 = 419.898$ | $D_2 = -38.356$ | $F_2 = -180.136$ |
| $B_3 = -0.367$ | $C_3 = -932.104$ | $D_3 = 129.099$ | $F_3 = 596.753$ |
| $B_4 = 0.500$ | $C_4 = 1076.587$ | $D_4 = -182.413$ | $F_4 = -835.176$ |
| $B_5 = -0.262$ | $C_5 = -506.806$ | $D_5 = 93.327$ | $F_5 = 425.388$ |

Therefore, the high-temperature flow stress of the alloy can be expressed by the constitutive equation with $Z$ parameters as:

$$\begin{cases} \sigma = \dfrac{1}{f(\varepsilon)} \ln\left\{ \left(\dfrac{Z}{e^{j(\varepsilon)}}\right)^{1/g(\varepsilon)} + \left[ \left(\dfrac{Z}{e^{j(\varepsilon)}}\right)^{2/g(\varepsilon)} + 1 \right]^{1/2} \right\} \\ Z = \dot{\varepsilon} \exp\left(\dfrac{g(\varepsilon)}{RT}\right) \end{cases} \tag{13}$$

where $f(\varepsilon)$, $g(\varepsilon)$, $h(\varepsilon)$ and $j(\varepsilon)$ represent the fifth-order strain polynomial functions of $\alpha$, $n$, $Q$, and $\ln A$ at different temperatures and strain rates, respectively. A constitutive relationship between stress and temperature, strain rate, and strain can be achieved, which is of great significance for predicting the hot deformation behavior and the simulation of the alloys.

### 3.3. Error Analysis

To test and verify the effectiveness of the established Arrhenius model in Equation (13), Part of the data can be extracted to test the accuracy of the Arrhenius model, the experimental stress value and the predicted stress value of temperature 400 °C and strain rate $0.001 \text{ s}^{-1}$ are compared and analyzed, as shown in Figure 5.

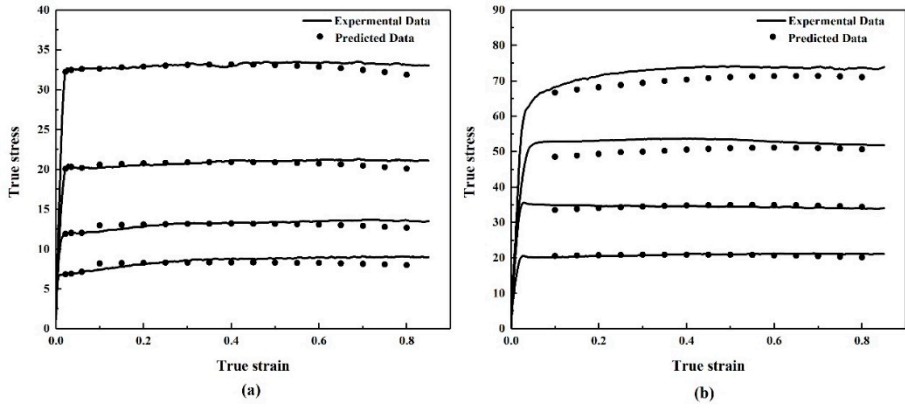

**Figure 5.** Comparison of Arrhenius model experimental and predicted values: (**a**) 0.001 s$^{-1}$ (**b**) 400 °C.

To further evaluate the accuracy of the model, the correlation coefficient ($R$) and the average absolute error ($\Delta$) are used to assess the accuracy of the model [25,26]. The $R$ and $\Delta$ can be expressed as:

$$R = \frac{\sum\limits_{i=1}^{N} \left(\sigma^i - \overline{\sigma}\right) \left(\sigma_p^i - \overline{\sigma}_p\right)}{\sqrt{\sum\limits_{i=1}^{N} \left(\sigma^i - \overline{\sigma}\right)^2 \sum\limits_{i=1}^{N} \left(\sigma_p^i - \overline{\sigma}_p\right)^2}} \tag{14}$$

$$\Delta = \frac{1}{N} \sum\limits_{i=1}^{N} \left| \frac{\sigma_{\exp}^i - \sigma_p^i}{\sigma_{\exp}^i} \right| \times 100\% \tag{15}$$

where $\sigma$ (MPa) is the experimental value, in thermal deformation, $\sigma_p$ is the predicted value, and $N$ is the sample size, $\bar{\sigma}$ and $\bar{\sigma}_p$ are their mean values.

The calculation results show in Figure 6 that the correlation coefficient of the Arrhenius model is $R = 0.9986$. The average absolute error is 2.89%. Therefore, the Arrhenius model obtained can effectively forecast the hot deformation behavior of the alloy, which is conducive to the simulation analysis of the alloy.

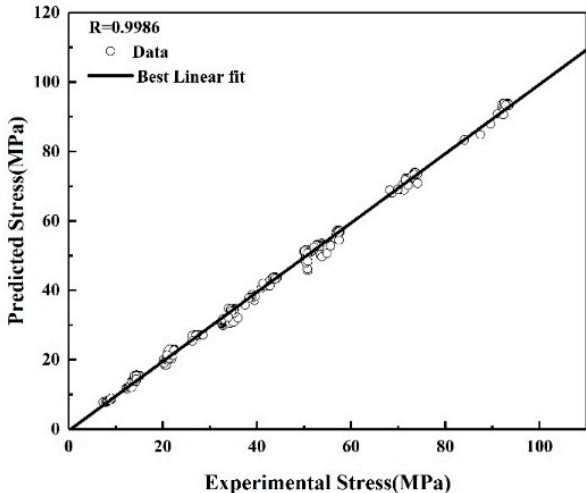

**Figure 6.** Correlation coefficient ($R$) and mean absolute error ($\Delta$) of constitutive equation of constitutive equation.

### 3.4. Processing Map and Activation Energy Map

On the basis of the dynamic material model (DMM) [27], the hot-processing map relates the mechanics of material deformation medium with power dissipation and the evolution of the alloy's microstructure. It can precisely control the processing parameters of materials and give products excellent quality and performance [28,29]. However, hot processing does not consider the difficulty of alloy deformation. It is not enough to study the hot workability only considering the power consumption efficiency $\eta$. On the contrary, hot workability should be considered from two aspects: deformation stability and deformation difficulty. The stable flow deformation region does not naturally indicate easy deformation. Activation energy characterizes the difficulty of the deformation process, because it reflects the ability of atoms to overcome energy barriers. Hence, through the coupling of the hot-processing map and AEP map, an optimal processing parameter is obtained.

Given strain temperature and deformation rate, the dynamic reaction constitutive equation of materials during deformation can be expressed as follows:

$$\sigma = K\dot{\varepsilon}^{m} \tag{16}$$

where $K$ is the material constant, and $m$ is the strain rate sensitivity index.

Figure 7 is a schematic diagram of Equation (17), and the rectangle represents the total energy $P$ absorbed by the material during the deformation process, which can be divided into $G$ and $J$ parts. $G$ represents the energy required for plastic deformation, most of which is released in the form of heat; $J$ represents the energy dissipated by the transformation of the microstructure. The total energy p absorbed by the thermally deformable member can be expressed as follows [30,31]:

$$P = \sigma\dot{\varepsilon} = G + J = \int_0^{\dot{\varepsilon}} \sigma d\dot{\varepsilon} + \int_0^{\sigma} \dot{\varepsilon} d\sigma \tag{17}$$

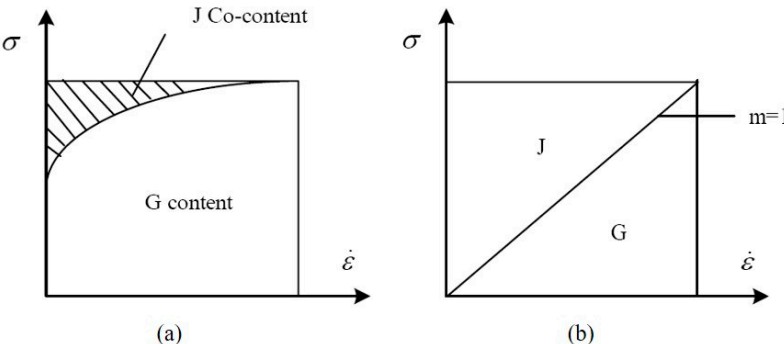

**Figure 7.** Dynamic constitutive equation for (**a**) given temperature and true strain (**b**) ideal linear dissipater.

Therefore, the strain rate sensitivity index $m$ can be expressed as follows:

$$m = \frac{dJ}{dG} = \left[ \frac{d(\ln \sigma)}{d\left(\ln \dot{\varepsilon}\right)} \right]_{\varepsilon, T} \tag{18}$$

The value of $\eta$ can be obtained from the value of $m$, the power dissipation coefficient $\eta$ can be expressed as follows:

$$\eta = \frac{2m}{m+1} \tag{19}$$

Prasad put forward the criterion of continuous instability based on the DMM model, which is based on the principle of an irreversible thermodynamic extreme. According to Prasad instability criterion, the instability parameter $\xi$ can be expressed as follows [32]

$$\xi\left(\dot{\varepsilon}\right) = \frac{d \ln\left(\frac{m}{m+1}\right)}{d \ln \dot{\varepsilon}} + m \leq 0 \tag{20}$$

The crucial parameters involved in the process of establishing the hot processing map mainly include the strain rate sensitivity index $m$, the power dissipation coefficient $\eta$ and the instability parameter $\xi$, among which the $\eta$ and $\xi$ is a dimensionless parameter, $\eta$ characterizes the power dissipation during the microstructure transformation of materials during hot working, $\xi$ is used to characterize the large rheological instability criterion. Figure 8 shows the 3D variation diagram of $\eta$ and $\xi$ with temperature and strain rate when the strain is 0.6. Figure 8a illustrates when the strain rate is constant, the power dissipation coefficient $\eta$ gradually increases with the increase of temperature, simultaneously when the temperature is constant, the power dissipation coefficient first rises and then declines with the addition of strain rate, and reaches the maximum when the strain rate is $0.1 \text{ s}^{-1}$. The value of $\eta$ reflects the process ability of the material, but it is not as if the larger the value, the better the processing performance, because the power dissipation coefficient may also be high in the processing instability area. Figure 8b illustrates, with the increase of temperature and the decrease of strain rate, that the instability parameter $\xi$ tends to increase gradually. As a matter of fact, given that the deformation mechanism of the alloy is complicated, it is essential to determine the best processing area and unsuitable processing area based on the comprehensive consideration of strain rate sensitivity, power dissipation and instability criterion.

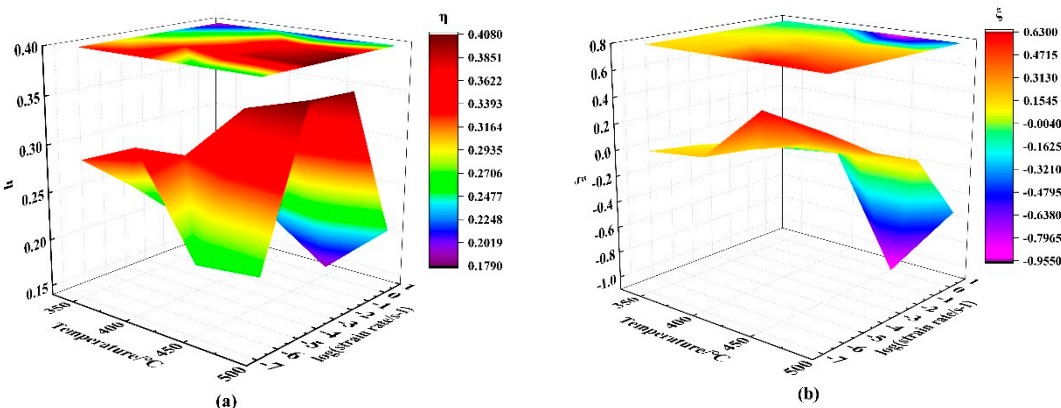

**Figure 8.** 3D variation diagram of $\eta$ and $\xi$ with temperature and strain rate at strain 0.6: (**a**) $\eta$ (**b**) $\xi$.

The hot-processing map can be acquired by superposition of power dissipation map and instability map. The hot-processing map of 0.6 strain and 0.8 strain of alloy are demonstrated in Figure 9. The contour lines of the figure represent the $\eta$, and the black area represents the $\xi$. When the strain is 0.6, there is a blue transparent instability area of region A as seen in Figure 9a, which is an unsuitable processing area. The dissipation coefficient $\eta$ in the B area is high and there is no instability area, which is the optimal processing area. However, there is no instability region in the C region, the dissipation factor is low, and the performance and quality of the processed product are not as good as the B region. When the strain is 0.8, there is a black instability area in area A as seen in Figure 9b, the dissipation coefficient of region B is higher than other regions, and the dissipation coefficient of area C is lower. Therefore, area B is the most suitable processing area. In summary, the stability processing area of the alloy is 460–500 °C (0.01–0.08 s$^{-1}$).

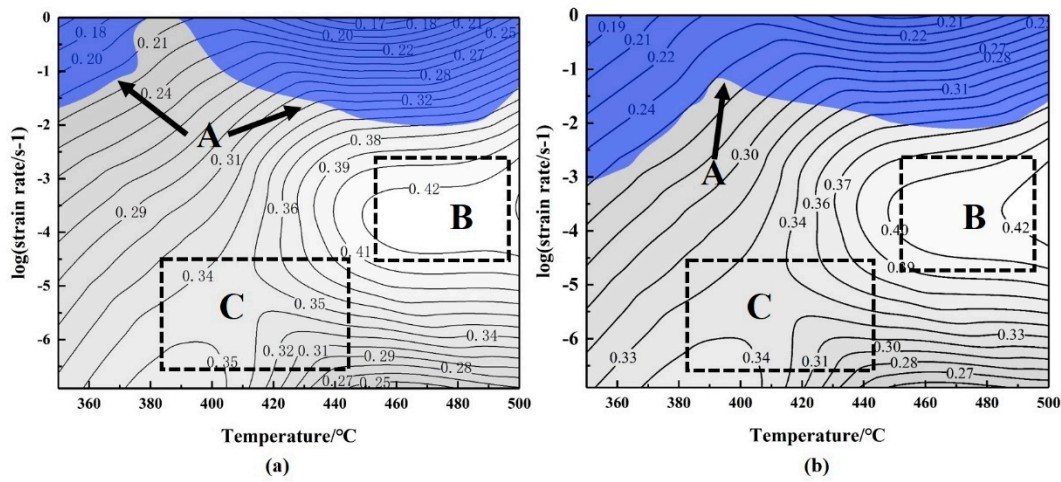

**Figure 9.** Hot processing map of Al–Zn–Mg alloy under different strains: (**a**) 0.6 (**b**) 0.8.

It is well known that the Z parameter is used to characterize the relationship between strain rate, deformation temperature and flow stress. Many studies have shown that the Z parameter can reflect the type of softening mechanism [33,34]. The lower lnZ value can promote the process of nucleation and growth of dynamic recrystallization (DRX) grains. In contrast, higher lnZ value increases the rate of dislocation accumulation and rearrangement, thus accelerating the process of dynamic recovery (DRV). Hence, the softening mechanism transforms from DRV under high Z value to DRX under low Z value. Dynamic recrystallization is beneficial to the hot-deformation process because it can generate stable rheology, and the area where dynamic recrystallization is often selected

to optimize the hot-forming process and control the microstructure. The data on the contour line in Figure 10a,b represent the power dissipation efficiency $\eta$. The lnZ parameter is represented by color in the figures (see the color bar on the right). It can be obtained that the value of lnZ parameter gradually decreases with the change from Domain I to Domain III, the softening mechanism transforms from DRV to DRX under, and this is beneficial to the hot forming of alloys. As shown in Figure 10a, it can be obtained that the minimum lnZ value is obtained near 500 °C/0.001 s$^{-1}$ and 500 °C/1 s$^{-1}$ at 0.6 strain. However, as shown in Figure 10b, the lnZ value increases near 500 °C/1 s$^{-1}$ at 0.8 strain, and this is because insufficient DRX due to excessive strain rate as the strain increases.

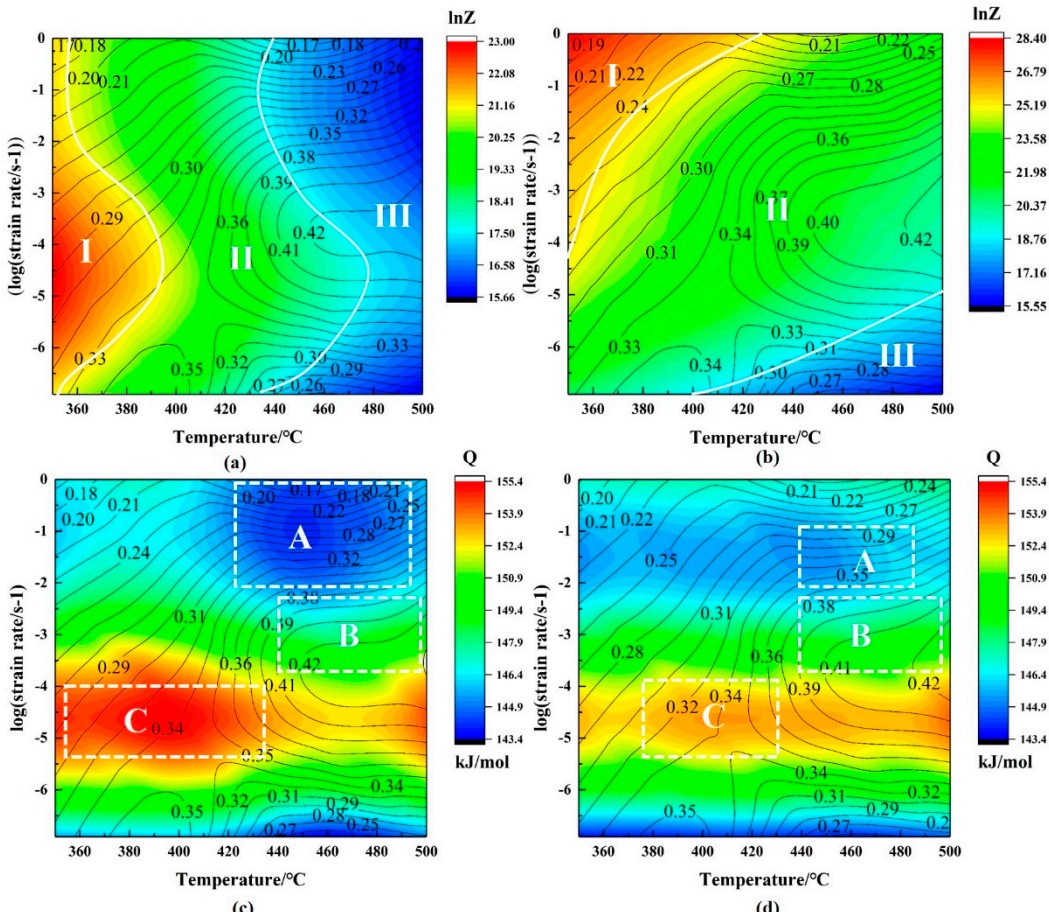

**Figure 10.** Z parameter maps and activation energy processing (AEP) maps at different strains of (**a,c**) 0.6, (**b,d**) 0.8.

The data on the contour line in Figure 10c,d represent the power dissipation efficiency $\eta$. Activation energy (AEP) value $Q$ is represented by color in the figures (see the color bar on the right). The higher $Q$ value under the condition of strain rate of 0.1. For metallic materials, a higher $Q$ represents a more difficult process of hot deformation, which indicates that the optimal processing area obtained from the hot-processing map needs to be further optimized, The $Q$ value in the A area is the lowest, and the $\eta$ value is 38%. However, the contour lines in this A area are closely distributed, which means that small changes in process parameters have significant effects on the g value, so this domain is not suitable for hot forming [35,36]. The $Q$ value of the B area is higher than that of the A area, but the contour line is sparse and the $\eta$ value is up to 42%. The $\eta$-range from 30% to 50% is usually attributed to the DRX, This implies that the deformation in area B is more stable. The $Q$ value of area C is the highest, which is not conducive to the thermal deformation of the alloy. In summary, the most suitable processing region is area B. After further optimization of the AEP map, the optimal processing area of the alloy is 460–500 °C (0.03–0.08 s$^{-1}$).

## 4. Discussion

### 4.1. Microstructure Evolution

As a high stacking fault energy (SFE) metal, the dynamic softening of Al–Mg–Zn alloy is primary contributed by dynamic recovery (DRV) rather than dynamic recrystallization (DRX). However, during the hot deformation of the alloys, as the deformation parameters change, dynamic recrystallization gradually occurs. In the meantime, discontinuous dynamic recrystallization (CDRX) is the main mechanism for recrystallization, the subgrain absorption dislocations generated by dynamic recovery enter the boundary, and the adjacent subgrains continuously rotate, migrate and merge to produce an orientation difference with the initial grains.

EBSD was used to analyze the microstructure of the specimen center perpendicular to the compression axis taken under the acceleration voltage of 20 KV and step size of 4.5 μm. The analysis results are listed in the Figure 11. In the EBSD maps, It can be noted that a low-angle boundary (LAGBs) with an angle between 2° and 15° is described as a thin line, and a high-angle boundary (HAGBs) than 15° is described as a thick line. Figure 11 shows the IPF diagram of EBSD of Al–6.32Zn–2.10Mg alloy under different deformation conditions. As shown in Figure 11a, the initial grains of the alloy are elongated perpendicular to the compression direction to form fibrous grains. A large number of subgrains with LAGBs was formed in the alloy at 350 °C/0.001 s$^{-1}$, which shows typical DRV microstructure characteristics. As shown in Figure 11c, many LAGBs were formed inside the deformed grains by sub-grain rotation evolution at 500 °C/0.001 s$^{-1}$, and some tiny DRX grains are formed near the initial grain boundary. At the same time, with the increases of deformation temperature, the deformed grains show an orientation gradient, which is attributed to the formation of substructures and CDRX. As in Figure 11d, the EBSD map of the alloy indicates that there are a large number of subgrains in the initial grain. This is because time is not enough for these substructures to rotate to form dynamic recrystallized grains as the strain rate becomes 1 s$^{-1}$. However, It can provide more energy storage and nucleation areas for subsequent dynamic recrystallization.

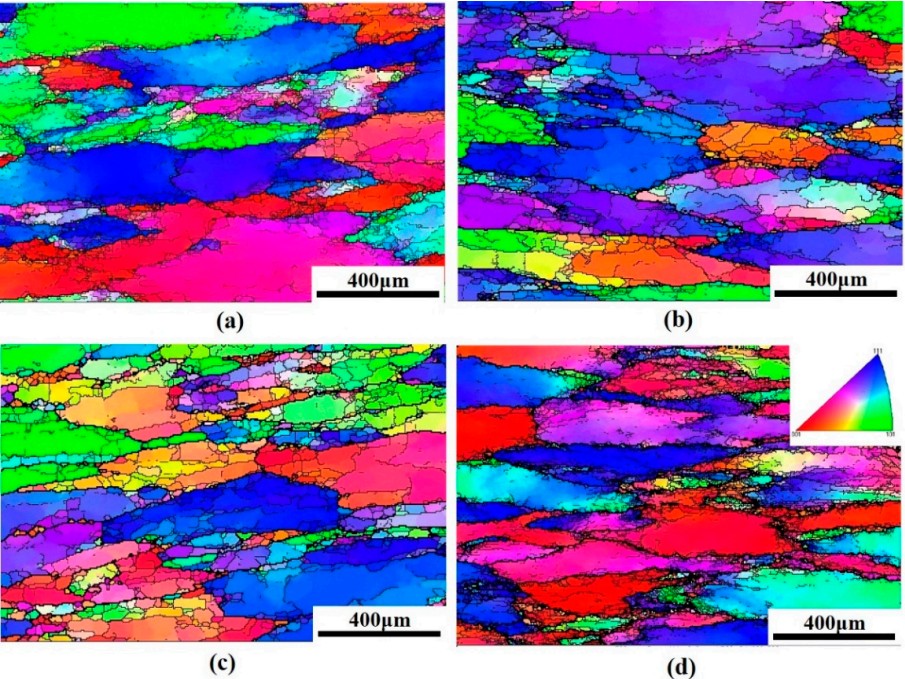

**Figure 11.** Electron backscatter diffraction (EBSD) maps of isothermal compression at different deformation temperatures and strain rates: (**a**) 350 °C/0.001 s$^{-1}$ (**b**) 450 °C/0.001 s$^{-1}$ (**c**) 500 °C/0.001 s$^{-1}$ (**d**) 500 °C/1 s$^{-1}$.

Figure 12 shows the DRX percentage maps at different deformation temperatures and strain rates. Comparing diagrams Figure 12a,c, as can be seen, as the temperature increases the proportion of dynamic recrystallization changes from 6.84% to 63.12%. This is because with higher temperatures, higher stored energy is generated, which spurs the grain boundaries' migration and promotes the nucleation of DRX. Comparing diagrams Figure 12c,d, as the temperature increases, the proportion of dynamic recrystallization changes from 41.76% to 12.49%. This is because at high strain rates, the time for grain boundary migration and sub-grain rotation is shortened, which is detrimental to DRX. This is in line with the previous statement. The lnZ value at 500 °C/0.001 s$^{-1}$ is the lowest, so the microstructure characteristic is the highest degree of DRX. At the same time, the DRX degree of 500/1 s$^{-1}$ is greater than the DRX degree of 300 °C/0.001 s$^{-1}$, which is consistent with the description of the lnZ parameter processing maps in Figure 10. However, as shown in Figure 12d, there is a large amount of deformed structure under the 500/1 s$^{-1}$ deformation condition, which is not conducive to hot processing. This also shows that a lot of energy is stored at this time, and further microstructure evolution is prone to occur. This is consistent with the 500 °C/0.01 s$^{-1}$ described in Figure 10, which has a lower $Q$ value and a lower $\eta$ value. Therefore, all these results indicate good consistency between the processing map and the observation of the microstructure.

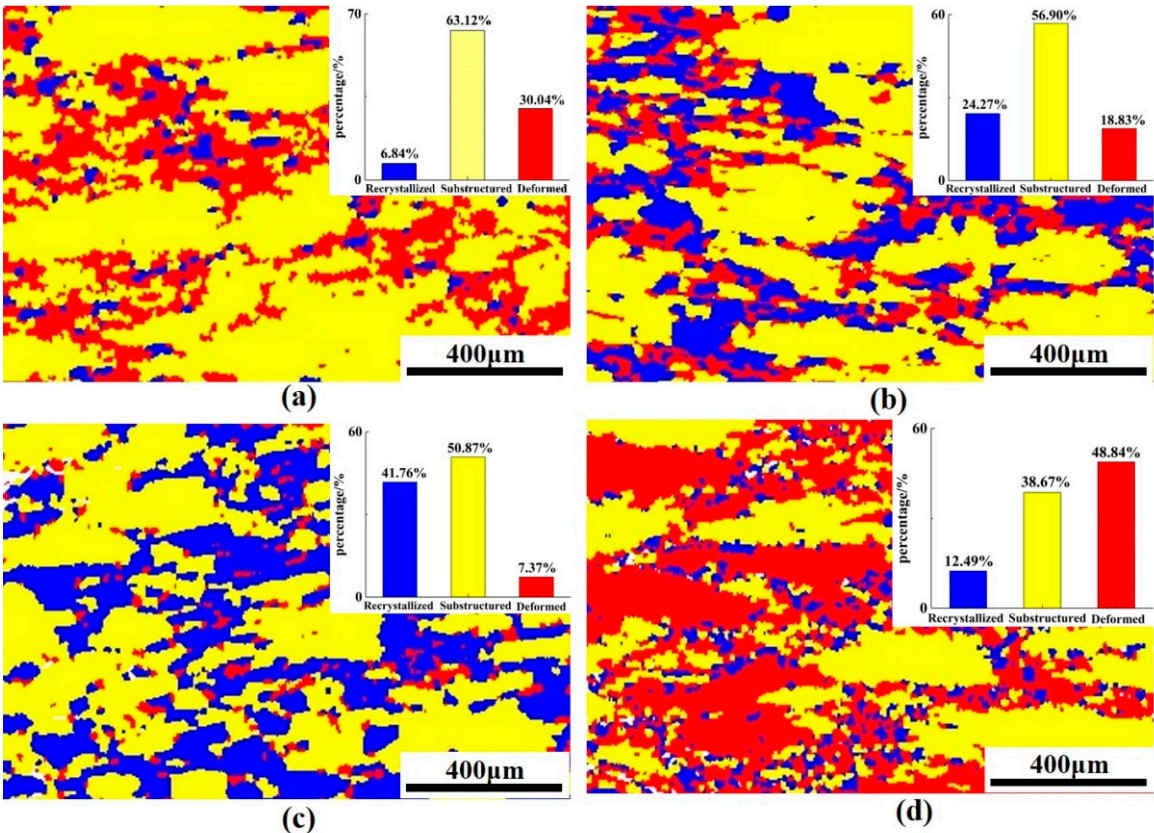

**Figure 12.** Dynamic recrystallization (DRX) percentage maps at different deformation temperatures and strain rates: (**a**) 350 °C/0.001 s$^{-1}$ (**b**) 450 °C/0.001 s$^{-1}$ (**c**) 500 °C/0.001 s$^{-1}$ (**d**) 500 °C/1 s$^{-1}$.

### 4.2. Dynamic Recrystallization in the Studied Alloys

In an assay to investigate the dynamic recrystallization characteristics in the Al–Mg–Zn series alloys, an enlarged EBSD IPF map was obtained. Generally, the mechanism of CDRX is to dynamically recover the subgrain with an orientation difference of less than 10° formed in the initial grains, and then gradually transform into large-angle grain boundaries through continuous subgrains' rotation through atomic diffusion. As shown in Figure 13a, obvious

subgrains can be observed in the red dotted part., showing a typical DRV microstructure. At the same time, as shown in Figure 13b, there are many LAGBs (misorientation angles between 2° and 15°) along the arrow direction, which further proves the existence of DRV. The cumulative misorientation along the arrow direction exceeds 15°, which indicates that gradual subgrains rotation, migration and coalescence have occurred. However, as shown in Figure 13c, CDRX grains are generated inside the deformed grain matrix, and the fact that very few necklace crystals formed along the initial grain boundary in the red dotted part is an obvious discontinuous dynamic recrystallization (DDRX) microstructure. As shown in Figure 13d, this misorientation map shows that there are some large-angle states, which are formed by the mechanism of CDRX. Therefore, CDRX is the main recrystalliza-tion mode in Al–Zn–Mg alloy, and DDRX is produced when the strain is large enough, without the obvious geometric dynamic recrystallization (GDRX) phenomenon.

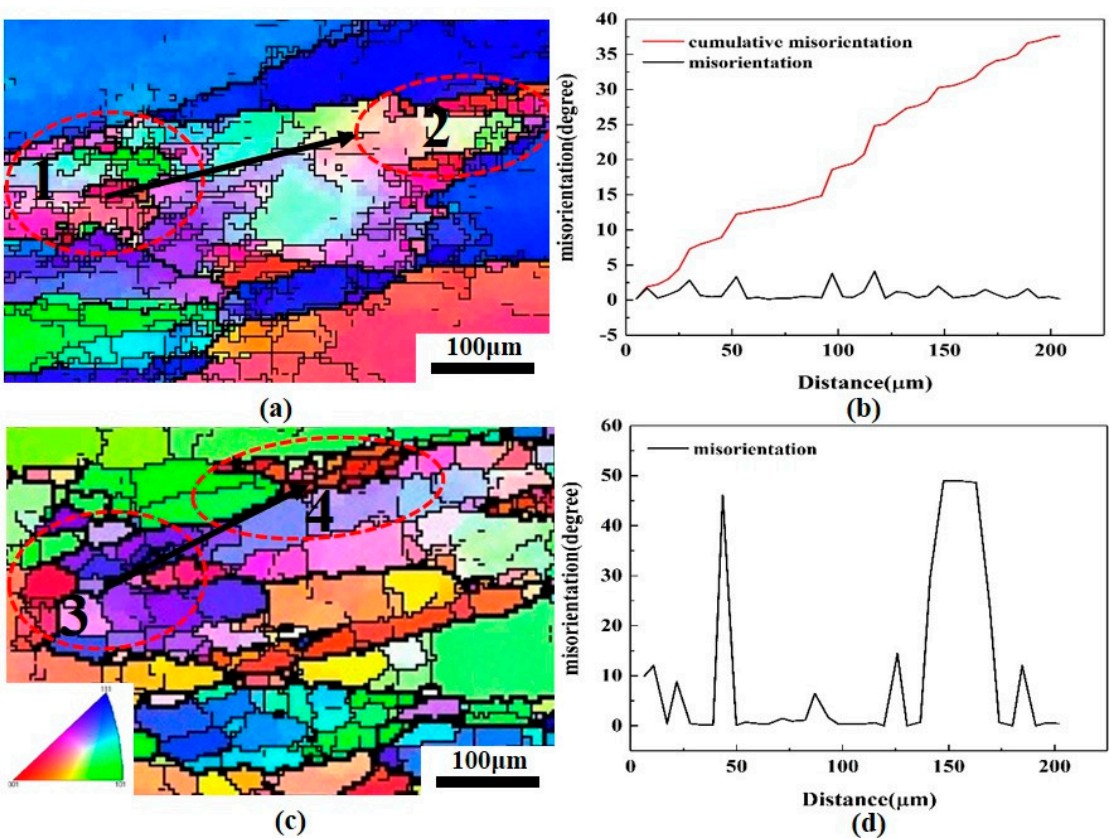

**Figure 13.** Enlarged EBSD IPF map and misorientation map: (**a,b**) 350 °C/0.001 s$^{-1}$ (**c,d**) 500 °C/0.001 s$^{-1.}$

Figure 14a,b are the kernel average misorientation (KAM) map and grain boundaries (GBs) map of the Al–Zn–Mg alloy at 350 °C/0.001 s$^{-1}$. The KAM maps can be directly interpreted according to the dislocation density that there is a positive correlation between KAM value and dislocation density [37,38]. For the alloy at 350 °C/0.001 s$^{-1}$, section regions within the initial grains have higher KAM values, which indicated the tangled dislocations were more concentrated within the initial grains in the alloy at a lower tem-perature. As shown in Figure 14b, some LAGB was observed inside the deformed grains of the alloy, and almost no fine DRX grains were formed. Therefore, the main softening mechanism of the alloy at lower deformation temperature was DRV, while CDRX was rarely observed. Figure 14c,d show the KAM map and GBs map of the Al–Zn–Mg alloy at 500 °C/0.001 s$^{-1}$. Dislocations were also formed along the initial grain boundaries of the deformed structure, and some LAGBs and tiny DRX grains were observed near the initial grain boundaries. In this condition, the subgrains along the initial grain boundary gradually rotate to form CDRX, and the formation characteristics of dislocations near the

initial grain boundary are observed correspondingly in the KAM map. The dislocations are uniformly formed along the initial grain boundary, and some LAGBs and DRX grains also formed near the initial grain boundary; this phenomenon shows the occurrence of DDRX. Meanwhile, a higher KAM value was obtained in partially deformed grains, and many LAGBs were observed accordingly, which indicated the existence of DRV, in this case, and DRV and DRX exist simultaneously in the Al–Zn–Mg alloy.

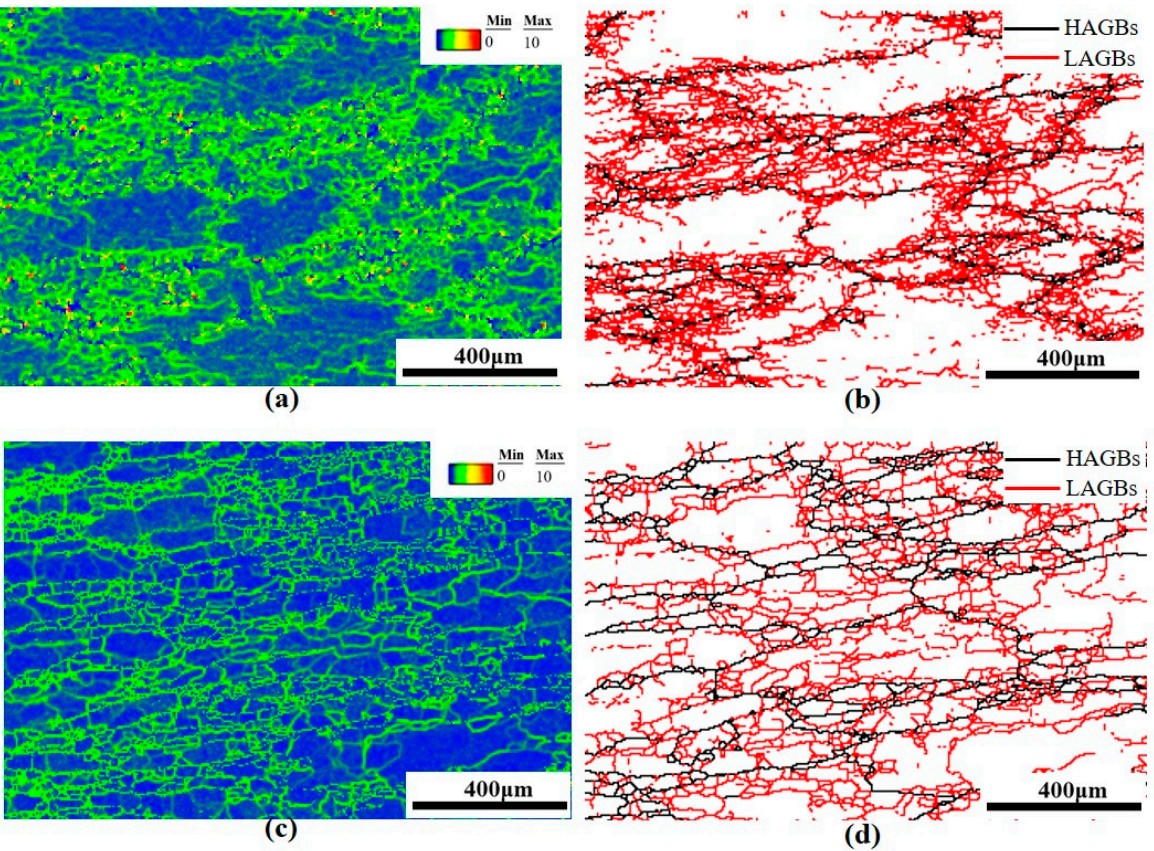

**Figure 14.** Kernel average misorientation (KAM) maps and grain boundaries' (GB) maps of the alloy at different deformation parameters: (**a**) (**b**) 350 °C/0.001 s$^{-1}$ (**c**) (**d**) 500 °C/0.001 s$^{-1}$.

## 5. Conclusions

In this research, the Al–6.23Zn–2.10Mg alloy has been subjected to an isothermal compression test to study its hot deformation behavior. The main results can be summarized as follows:

1.  The true stress increased significantly with increasing strain rate and decreasing deformation temperature. The modified Arrhenius constitutive equation based on hyperbolic sine function was established. The correlation coefficient of the Arrhenius model was R = 0.9986 and the average absolute error was 2.89%, which can accurately describe the high-temperature flow stress.
2.  The hot-processing map was established based on DMM, and the stability processing area for the hot-processing map was 460–500 °C/0.01–0.08 s$^{-1}$. Then, the Z parameter processing map and activation energy processing maps were established for further optimization. Eventually, the optimal processing parameters of the alloy were 460–500 °C (0.03–0.08 s$^{-1}$).
3.  EBSD techniques were used to analyze the evolution of the grain structure during the deformation process. Based on the findings, the reasonability of the AEP map and Z parameter map was verified. Meanwhile, in the Al–Zn–Mg alloy, dynamic recovery occurs first and subgrains with an orientation difference of less than 10°

are formed in the initial grains. The subgrains are gradually converted to large-angle grain boundaries through constantly absorbing dislocations and continuous subgrains' rotation, showing typical CDRX microstructure characteristics. Then with the continuous increase of strain, the strain induces migration of large-angle grain boundaries to produce DDRX, forming necklace grains along the initial grain boundary.

**Author Contributions:** Conceptualization, Q.W.; literature search, Q.W.; figures, Q.W.; investigation, Q.W. and M.L.; data collection Q.W.; data analysis, Q.W.; data interpretation, Q.W.; writing, Q.W.; funding acquisition, Z.X. project administration, Z.X. and Y.H.; resources, Y.H.; supervision, Y.H.; writing—review & editing, J.H. All authors have read and agreed to the published version of the manuscript.

**Funding:** This research was funded by the Fundamental Research Funds for the Central Universities of Central South University (Project Number: 1053320191914) and Hunan Provincial Innovation Foundation for Postgraduate (Project Number: 150110001).

**Institutional Review Board Statement:** Not applicable.

**Informed Consent Statement:** Not applicable.

**Data Availability Statement:** Data is contained within the article.

**Acknowledgments:** Authors are grateful for the support.

**Conflicts of Interest:** The authors declare no conflict of interest.

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
