# Peer review of "Hot Deformation Characteristics and Processing Parameter Optimization of Al–6.32Zn–2.10Mg Alloy Using Constitutive Equation and Processing Map"

_metals, doi:10.3390/met11020360_

Round 1
Reviewer 1 Report
Dear Authors,
although Your research sounds interesting, please first review the text in full detail and check English grammar.
In Chapter 2, after Table 1, You mention that You kept the specimen for 5 min on mentioned temperature, to ensure that the internal temp. of the specimen is uniform. Are You sure that 5 min is enough to establish uniform temperature across the specimen? Also, the sentence following previous statement ("The compression reduction for the specimens set as 60%, i.e., true strain was about 0.9.") is somehow unclear and further explanation is needed.
In Chapter 3, stress - strain curves are given. How did You obtain stress values?
In addition, I believe that there are many research papers published in Metals Journal that cover given topics, so additional background and reference check is advised.
Best regards.
Author Response
Dear Reviewer:
Thank you for your comments concerning our manuscript entitled “To uncover the hot deformation characteristics of Al-6.32Zn-2.10Mg alloy using constitutive equation and processing map” (ID: metals-1085236). Those comments are all valuable and very helpful for revising and improving our paper, as well as the important guiding significance to our researches. We have studied comments carefully and have made correction which we hope meet with approval. Please see the attachment for specific responses!
Best regards.

Reviewer 2 Report
Notes in the file for authors.

Author Response

(The authors gave the same response as above.)

Reviewer 3 Report
Dear authors of the mannuscript,
I find your research well thought out and concise. However, below are some suggestions for improving the manuscript.
General remarks:
English language and style would benefit from some editing. Some words are not used correctly, some sentences are too long, there are some problems with grammar... but as a reviewer, I did not concentrate on the english language, as this should be done by someone qualified in scientific writting in English language. However, the manuscript is legible and fully understandable as-is, so just a check of grammar / some editing should be sufficient.
Introduction:
The Introduction sections gives reasonable background on the Al-Zn-Mg alloy processing maps and previous research, including references to other available publications. English needs some modifications and a good spell check should be ran through the manuscript.
The aims of the research are also clearly stated, but justification for this particular composition selection is missing. I agree that there is no hot processing map available for this particular alloy composition, but processing maps are available for similar compositions/alloys. So, what is really special to choose this particular composition? This needs to be improved.
Experimental:
Missing are the details of the EBSD set-up (however, it is mentioned in the EBSD section). Otherwise, the experimental section is sufficiently clear and would allow another researcher to repeat the experiment.
Results:
Fig 2: The x-axis label needs to be fixed (strain, not starin).
The last paragraph on page 4 (To work out N ... ) is hard to understand, so some rewritting is suggested. Also, way too many decimal places when reporting the results.
Pg 5: Inroduce Eq(11)... - it is not Eq 11 that is inserted into Eq3 to get to Eq 10. Please re-check the referenced Equations, so that it is clear (also valid for other equations).
Last paragraph on pg5: (Remarkably, it may be inaccurate .....) -I do not understand even after reading a couple of times. Please rewrite and check for clarity.
Fig4: the axes labels are too small and hard to read.
Fig8: since hot processing map is the focus of this article, the image should be larger.
Discussion:
Accelerating voltage was most probably 20kV and not 20V. Also, step size is missing units (I suppose it is micrometers?) Did you do any post-processing (noise reduction etc) for the EBSD maps? How did you prepare the samples (and what is approximate hit rate?)
Fig11, KAM maps: missing a legend Also, it would be better to just show KAM (and maybe IQ superimposed) and not the grain boundaries (on KAM maps). This way it is better visible, with no black lines. Also, the scale bar is dark and cannot be seen. Also, in GB maps, show some legend for people who will just look at the images.
Fig12: Which EBSD map do these plots refer to? What was the analysed area?
Conclusions:
Well summarised and concise, makes sense.
In summary, the manuscript is interesting and the research design is good. However, language must be improved (so that it is clearer), and some suggestions for improving data interpretation are also given. After modifications, the manuscript is suitable for publication in my opinion.
Author Response

(The authors gave the same response as above.)

Reviewer 4 Report
1) In my opinion, the article does not contain significant factual errors, but its novelty and contribution to scientific knowledge seems to be low. The part dealing with hot flow stress uses the very standard methodology (activation energy, polynomial regression) as well as many other similar papers, only applied to another alloy. Much more interesting is the part focused on the microstructure characteristics (EBSD).
2) Occasionally, trivial / banal messages are presented, almost inappropriate for the renowned journal Metals (Q1), see e.g.
- ... the flow stresses increase with the increasing of strain rate or the decreasing of deformation temperature (in Abstract);
- The lnZ gradually decreases with increasing temperature and decreasing strain rate [30]. (in Discussion).
3) The abstract is quite inappropriately formulated - it should briefly describe the results achieved, not only the experiments performed.
4) The Discussion deals only with structural analyzes (and dynamic softening processes) but completely ignores the results related to the flow stress model. It is not enough to characterize the accuracy of the developed flow stress model (in Results), but also to compare the results with other authors (hot deformation activation energy, flow curves) - these are only briefly mentioned in the Introduction.
5) The legitimacy of the description of flow curves by a uniform model should be justified, when the stress-strain curves are obviously the result of various softening processes (recovery, recrystallization). Of course, the accuracy of the phenomenological model is good because all flow curves show a very flat course.
6) From the point of view of fundamental research, the novelty of the obtained results is controversial. If the research was to give practical results (see the first two sentences in the Introduction), their applicability should be described in more detail.
Author Response
Dear Reviewers:
Thank you for your comments concerning our manuscript entitled “To uncover the hot deformation characteristics of Al-6.32Zn-2.10Mg alloy using constitutive equation and processing map” (ID: metals-1085236). Those comments are all valuable and very helpful for revising and improving our paper, as well as the important guiding significance to our researches. We have studied comments carefully and have made correction which we hope meet with approval. Please see the attachment for specific responses!
Best regards.

Reviewer 5 Report
(1) The submitted manuscript applies well-known approaches to evaluate a hot deformation behavior of the Al-Zn-Mg alloy. On the one hand, the modified Arrhenius relationship and conventional processing maps can be considered to be enough to gain information about a hot deformation behavior of different alloys. On the other hand, they bring no new point of view or added value. Authors should explain why they applied these approaches, and at least cite some newer approaches for the processing map evaluation – see e.g.:
[a] Characterization of hot workability of 5052 aluminum alloy based on activation energy-processing map. https://doi.org/10.1007/s11665-019-04367-7
[b] Correlation among the Power Dissipation Efficiency, Flow Stress Course, and Activation Energy Evolution in Cr-Mo Low-Alloyed Steel. https://doi.org/10.3390/ma13163480
(2) It seems, there is a mistake in Figure 1 – deformation temperature: 300 – 450 °C. Is this temperature range correct? There should be maybe 350 – 500 °C.
(3) On the page 6, there is a reference to Eq. (14) in the first sentence. Maybe, there should be reference to Eq. (12).
(4) Equations for the parameters η and ξ should be introduced.
(5) Quality of Figure 7 should be increased – especially, the legend font is now practically unreadable.
Author Response
Dear Reviewer:
Thank you for your comments concerning our manuscript entitled “To uncover the hot deformation characteristics of Al-6.32Zn-2.10Mg alloy using constitutive equation and processing map” (ID: metals-1085236). Those comments are all valuable and very helpful for revising and improving our paper, as well as the important guiding significance to our researches. (The two articles you provided have been carefully read and a newer approaches for the processing map evaluation has been added to my article) We have studied comments carefully and have made correction which we hope meet with approval. Please see the attachment for specific responses!
Best regards.

Round 2
Reviewer 1 Report
Dear authors,
thank You for Your improvements of the text and for the explanations.
Reviewer 2 Report
Dear authors,
The comments I have submitted are only a form of discussion of the matter. They mainly result from a misunderstanding of the presented content. So you are for a sign that is what you wrote some reason it is incomprehensible to the reader. But it may also be the result of mistakes in your interpretation of the admittedly difficult issues. That is why the review of the person standing on the sidelines is so important for the quality of the description of the phenomena presented. I am glad that my comments found your approval. And the wide range of corrections you have introduced positively affects the quality of the article.
However, from the research point of view, there is still a noticeable lack of appropriate analyzes of the structure of the deformed material. Especially in the area indicated as optimal. The basic verification of the results obtained by you is a thorough analysis of the deformed structures. Especially in area B and C. This undoubtedly lowers the level of the article, but it is also only one of the presented aspects of the whole research.
What's next:
- …the Q = 147.805 KJ/mol … - it is a value derived from mathematical calculations and the number of decimal places can be much higher. In the measuring technique, their number provides information about the accuracy of the instrument, which is very important information. In the calculations, it is assumed that the result should be recorded with an accuracy of 2 significant places, i.e. 14 × 101 KJ/mol. And this is sufficient, because my experience shows that the value of this energy is usually between 100 and 400 KJ/mol. Let’s say that 148 KJ/mol will be better.
- Agreed, coefficient η defined by this formula is a unitless value. But as you wrote yourself, it is called as the efficiency of power dissipation. And efficiency is expressed as a percentage. Both variants are correct and adequate, but efficiency is efficiency.
By the way, thank you for quoting my article. It's a little past history. Please note that you're writing for the world, so do not limit your discernment to a close only researchers.
Good luck
Reviewer 4 Report
Authors reflected almost all of my suggestions and markedly improved the article.